# MRLocus: Identifying causal genes mediating a trait through Bayesian estimation of allelic heterogeneity

Anqi Zhu[1], Nana Matoba[2,3], Emma P. Wilson[2], Amanda L. Tapia[1], Yun Li[1,2,4], Joseph G. Ibrahim[1], Jason L. Stein[2,3], Michael I. Love[1,2]*

1 Department of Biostatistics, University of North Carolina at Chapel Hill, Chapel Hill, North Carolina, United States of America, 2 Department of Genetics, University of North Carolina at Chapel Hill, Chapel Hill, North Carolina, United States of America, 3 UNC Neuroscience Center, University of North Carolina at Chapel Hill, Chapel Hill, North Carolina, United States of America, 4 Department of Computer Science, University of North Carolina at Chapel Hill, Chapel Hill, North Carolina, United States of America

☯ These authors contributed equally to this work.
* michaelisaiahlove@gmail.com

**Data Availability Statement:** All real and simulated preprocessed data used to run the software in the article can be found at the GitHub repository: https://github.com/mikelove/mrlocusPaper eQTL

## Abstract

Expression quantitative trait loci (eQTL) studies are used to understand the regulatory function of non-coding genome-wide association study (GWAS) risk loci, but colocalization alone does not demonstrate a causal relationship of gene expression affecting a trait. Evidence for mediation, that perturbation of gene expression in a given tissue or developmental context will induce a change in the downstream GWAS trait, can be provided by two-sample Mendelian Randomization (MR). Here, we introduce a new statistical method, MRLocus, for Bayesian estimation of the gene-to-trait effect from eQTL and GWAS summary data for loci with evidence of allelic heterogeneity, that is, containing multiple causal variants. MRLocus makes use of a colocalization step applied to each nearly-LD-independent eQTL, followed by an MR analysis step across eQTLs. Additionally, our method involves estimation of the extent of allelic heterogeneity through a dispersion parameter, indicating variable mediation effects from each individual eQTL on the downstream trait. Our method is evaluated against other state-of-the-art methods for estimation of the gene-to-trait mediation effect, using an existing simulation framework. In simulation, MRLocus often has the highest accuracy among competing methods, and in each case provides more accurate estimation of uncertainty as assessed through interval coverage. MRLocus is then applied to five candidate causal genes for mediation of particular GWAS traits, where gene-to-trait effects are concordant with those previously reported. We find that MRLocus's estimation of the causal effect across eQTLs within a locus provides useful information for determining how perturbation of gene expression or individual regulatory elements will affect downstream traits. The MRLocus method is implemented as an R package available at https://mikelove.github.io/mrlocus.

data on tibial artery and liver was obtained from the Genotype-Tissue Expression (GTEx) project v8 (GTEx Consortium et al., 2017). Bernhard Weber, corresponding author on Strunz et al. 2018 can be contacted at bweb@klinik.uni-regensburg.de.

**Funding:** AZ and JGI were supported by NIH National Institute of General Medical Sciences R01 GM070335. Additional support to AZ was from NIH National Cancer Institute P01 CA142538. NM, ALT, JLS, and MIL were supported by NIH National Institute of Mental Health R01 MH118349. YL was supported by NIH National Heart, Lung, and Blood Institute R01 HL129132, NIH National Institute of General Medical Sciences R01 GM105785, and NIH National Institute of Child Health and Human Development U54 HD079124. Additional support to JLS was from NIH National Institute of Mental Health R01 MH121433 and R01 MH118349 (URL for the NIH funders:https://www.nih.gov/institutes-nih/list-nih-institutes-centers-offices). The funders had no role in study design, data collection and analysis, decision to publish, or preparation of the manuscript.

**Competing interests:** The authors have declared that no competing interests exist.

## Author summary

Genome-wide association studies (GWAS) have identified many loci associated with complex traits and diseases. Expression quantitative trait loci (eQTL) may help to explain mechanisms of GWAS associations, if the gene has a role as a mediator of the trait or disease. Loci that exhibit allelic heterogeneity, that is, loci containing multiple causal variants, offer the opportunity to investigate whether effects are concordant and proportional across eQTL and GWAS; if the gene is a partial mediator of the trait, the sign and size of the effects across distinct eQTL variants should be reflected in GWAS associations. Such a Mendelian Randomization (MR) analysis of individual loci is complicated by moderate sample sizes in eQTL studies and linkage disequilibrium (LD), resulting in complex patterns of estimated effect sizes for eQTL and GWAS. We develop a statistical model, MRLocus, with two steps: selection of eQTL SNPs to act as instruments in the MR analysis of a genetic locus, and estimation of the gene-to-trait mediation effect taking instrument uncertainty into account. In simulation, the method has higher accuracy and better uncertainty measures compared to other competing methods, and we compare its estimates on candidate causal gene-trait pairs from literature.

## Introduction

Genome-wide association studies (GWAS) have identified many loci associated with complex traits and diseases. A major goal now is to understand the mechanism by which non-coding genetic variation influences trait levels through changes in gene expression. This involves identifying the causal variants at a locus, determining if the same variants are associated with both gene expression and trait, and disambiguating mediation from pleiotropy [1]. Proposing mediating genes from existing expression quantitative trait loci (eQTL) and GWAS resources will lead to experiments that test whether modulating gene expression influences traits, and therefore inform further research and development of treatments.

Current efforts to identify the genes underlying GWAS risk often make use of either colocalization of GWAS signal with eQTLs, or expression imputation. In colocalization, statistical models are used to probabilistically assess if the same genetic variants within a locus are likely to be causally contributing to both eQTL and GWAS signals, taking into account the structure of linkage disequilibrium (LD) for a given population [2–8]. Expression imputation methods, as in transcriptome-wide association studies (TWAS), add additional information by including subthreshold signal for both GWAS and eQTL to identify which genes' expression may have a non-zero local genetic correlation with a given GWAS trait [9–11]. Further refinements of TWAS statistical models have allowed for probabilistic fine-mapping within loci harboring multiple candidate genes by accounting for LD structure, as in FOCUS [11].

Though colocalization and expression imputation suggest genes involved in a trait, neither method is designed to disambiguate between pleiotropy and mediation. The latest generation of methods for determining those genes involved in mediating GWAS signal have combined the approaches of colocalization and expression imputation with statistical techniques from the field of Mendelian randomization (MR) [12–14], as evaluated and reviewed recently [15,16]. Intuitively, these methods work by determining if those genetic variants which influence gene expression also influence a downstream trait in proportionate degrees. Evidence for mediation is provided by randomized genetic variation used to perturb gene expression and observing the expression effects propagated to traits. With access to genotype, expression, and trait data, classical mediation techniques can be employed, as in the methods CIT [17] and

SMUT [18], while MR-link [19] makes use of individual-level data from a GWAS study and summary statistics for eQTL to perform MR analysis.

Other methods testing gene-to-trait mediation require only summary statistics from eQTL and GWAS studies, as it is rarely possible to have access to the per-participant genotypes, expression values, and GWAS trait values [20]. Methods such as CaMMEL [21], TWMR [22], LDA-MR-Egger [23], PMR-Summary-Egger [24], PTWAS [25], MESC [1], MR-Robin [26] allow for estimation of gene-to-trait effects from eQTL and GWAS summary statistics, integrating across multiple SNPs within a locus. The ability to accurately estimate the gene-to-trait mediation effect within a locus relies on having multiple independent "instruments", SNPs which are found to be associated with the potential mediator (gene expression), and which plausibly only affect the downstream trait through the mediator. Across studies, between 29–50% of genes provide evidence for *allelic heterogeneity*, that is, having more than one independent cis-eQTL per gene, with estimates varying by tissue and study sample size [27–29]. Some genes provide evidence for up to 13 independent cis-eQTL signals, detected by conditional analysis in peripheral blood [28]. A recent study integrating neonatal gene expression with GWAS of auto-immune and allergic disease performed MR analysis across 52 genes that had three or more cis-eQTLs [30]. Therefore, while not all genes display allelic heterogeneity at current eQTL study sample sizes, it is common enough to allow for mediation modeling of many candidate genes.

Existing methods for assessing whether expression of a particular gene in some context may mediate GWAS signal have primarily focused on their ability to perform genome-wide mediation scans across multiple tissues or cell types. This is a critical task in determining the genetic architecture and the most relevant molecular contexts for a trait (e.g. tissues, cell types, or developmental stage) which are often not known a priori. However, when considering functional follow-up experiments at an individual locus, investigators may weigh accurate quantification of the *uncertainty* regarding a potential gene-to-trait effect, as well as the *heterogeneity* of gene-to-trait effects across signal clusters within a locus.

Here, we propose MRLocus, a Bayesian model for estimating the gene-to-trait effect from multiple independent signal clusters for one gene, as well as for estimating the heterogeneity of the effects across clusters. We have designed our method for prioritization of genes in functional experiments, where the genes under study have already been identified as candidates for mediation, having emerged from one of the global mediation scanning methods described above, or from colocalization or TWAS. MRLocus performs estimation of the gene-to-trait effect itself, as our focus is on experimental follow-up, as opposed to estimation of the percent of mediated heritability. In comparisons with other recently developed methods for identifying mediating genes from eQTL and GWAS summary data and LD matrices, TWMR and PTWAS, MRLocus was often more accurate in estimation of the gene-to-trait effect across simulated eQTL and GWAS experiments, and had higher and closer to nominal coverage of the true effect when considering its credible intervals. Using existing eQTL and GWAS data, we also estimate mediation effects at previously reported and experimentally validated loci. The MRLocus method is implemented as an open source R package with full function documentation and a software vignette demonstrating its use, publicly available at https://mikelove.github.io/mrlocus.

## Materials and methods

### MRLocus method

MRLocus consists of two steps, (1) colocalization and (2) MR slope fitting, each of which use Bayesian hierarchical models specified in the Stan probabilistic programming language, and with posterior inference using the Stan and RStan software packages [31]. As in PTWAS [25], MRLocus estimates the gene-to-trait effect or slope by identifying "nearly-LD-independent"

signal clusters and then assessing the strength of evidence of mediation and the heterogeneity of the allelic effects. We refer to "nearly-LD-independent" clusters meaning non-overlapping sets of SNPs with low LD: MRLocus first uses PLINK's clumping algorithm [32] with $r^2 < 0.1$ to identify signal clusters, sets of SNPs correlated with gene expression based on eQTL p-value (as discussed below), whereas PTWAS uses DAP [33] for identifying signal clusters. MRLocus additionally performs trimming of signal clusters to ensure $r^2 < 0.05$ for cluster representative SNPs, first using across-cluster $r^2$ based on index SNP, then later using the candidate causal SNP from the colocalization step. Here, "trimming clusters" refers to removing clusters, prioritizing those clusters with lower index SNP eQTL p-value. Alternatively, conditional analysis could be used as discussed in a recent coloc methods paper [34]. As with other gene mediation methods mentioned above, we focus here on common SNPs (using a minor allele frequency (MAF) filter on real and simulated data of 0.01). To the extent that a SNP or genetic variant gives rise to both the eQTL and GWAS signal and is in the set analyzed by MRLocus, then the model has a chance to find the "causal" SNP, although in general MRLocus may identify a SNP which is in high LD with the causal SNP.

In contrast to other methods for estimating gene-to-trait effects, MRLocus additionally performs a colocalization step prior to slope fitting, using eQTL and GWAS summary statistics (estimated coefficients and standard errors (SE)), based on LD matrices (either distinct matrices when eQTL/GWAS are performed in different populations, or a single shared matrix can be used when eQTL/GWAS are performed in the same population). The colocalization step attempts to identify a single candidate causal SNP per nearly-LD-independent signal cluster. Here "candidate causal SNP" refers to the hypothesis that the SNP gives rise to both the observed eQTL and GWAS signal, given the LD matrices. The colocalization step produces posterior estimates that assess the degree to which the summary statistics and LD matrices support the causal hypothesis per signal cluster (see S1 Methods for details on the statistical model). If the SNP is a strong candidate for causing both the eQTL and GWAS signal in a signal cluster, then the posterior estimates for the eQTL and GWAS effect sizes will be large (in absolute value) for the SNP, and near 0 for the other SNPs. If the SNP is a strong candidate for only the eQTL signal, but not for the GWAS signal, then the posterior estimate for the chosen SNP for the GWAS signal will be near 0. Finally, we note that prior to the colocalization step, MRLocus performs collapsing of highly correlated SNPs (threshold of 0.95 correlation), such that the final "per-SNP" results actually correspond to results for representatives from sets of highly correlated SNPs. MRLocus also performs allele flipping such that all SNPs are coded to be in positive LD correlation to the index SNP that has a positive estimated coefficient for eQTL (S1 Methods). This ensures that the statistical modeling and visualizations are always referring to the effect of an expression-increasing allele.

MRLocus's colocalization step is motivated by the eCAVIAR model [5] as it formulates a generative model for the summary statistics based on true underlying signals, but is distinct from the eCAVIAR model in two respects. First, eCAVIAR models the *z*-scores from eQTL and GWAS, while MRLocus directly models the estimated coefficients, as our focus is on estimation of the gene-to-trait effect, which can be conceived as in other MR applications as a regression of coefficients from GWAS on eQTL. eCAVIAR does not output posterior effect sizes for the two studies, which is the originally intended input to MRLocus slope estimation. Second, eCAVIAR uses a multivariate normal distribution to model the vector of observed *z*-scores in a locus, while MRLocus uses a univariate distribution to model the estimated coefficients of the SNPs in each nearly-LD-independent signal cluster. The univariate distribution was chosen for its increased performance in accuracy and in efficiency in model fitting, as well as for flexibility in specification of prior distributions. As the methods have distinct but related functionality, we also assessed the use of eCAVIAR in lieu of MRLocus's colocalization step, for choice of candidate causal SNPs to

provide to the subsequent slope fitting step (referred to here as "eCAVIAR-MRLocus"). As with MRLocus colocalization, clusters are trimmed such that across-cluster $r^2 < 0.05$ based on eCAVIAR chosen SNPs, but collapsing of highly correlated SNPs is not performed, as it is not required for eCAVIAR input. The SNP with largest colocalization posterior probability (CLPP) is chosen among those SNPs with p-value below the clumping threshold.

In its implementation of colocalization, MRLocus uses a horseshoe prior [35] on the true coefficients for eQTL and GWAS signal per signal cluster, which helps to induce sparsity in the posterior estimates of the coefficients prior to mediation analysis (S1 Methods). The proposed use of the horseshoe prior during colocalization to identify putative causal SNPs from eQTL and GWAS coefficients within a signal cluster is distinct from other Bayesian MR methods' use of the horseshoe prior on pleiotropic effects or on the mediation slope [36–38].

MRLocus's slope fitting step involves estimation of the gene-to-trait effect across signal clusters (Fig 1). For slope estimation, the best candidate SNP per nearly-LD-independent signal cluster is chosen, based on which SNP has the largest posterior mean of the eQTL effect size from the colocalization step. A hierarchical model is used to perform inference on parameters of interest, in this case the slope ($\alpha$) of true GWAS coefficients on true eQTL coefficients. In contrast to a typical inverse-variance weighted MR slope estimation, MRLocus additionally includes uncertainty on the effect size of the *instruments*, which we find leads to better coverage properties of credible intervals in simulated datasets. For study of putative mediator A (e.g. eQTL) and downstream trait B (e.g. GWAS), the following hierarchical model is fit using estimated coefficients $\hat{\beta}_j$ and standard errors $se_j$ across signal clusters $j \in 1,\ldots,J$:

$$\hat{\beta}_j^A \sim \mathrm{N}(\beta_j^A, se_j^A)$$

$$\hat{\beta}_j^B \sim \mathrm{N}(\beta_j^B, se_j^B)$$

$$\beta_j^A \sim \mathrm{N}(0, \mathrm{SD}_\beta)$$

$$\beta_j^B \sim \mathrm{N}(\alpha\beta_j^A, \sigma)$$

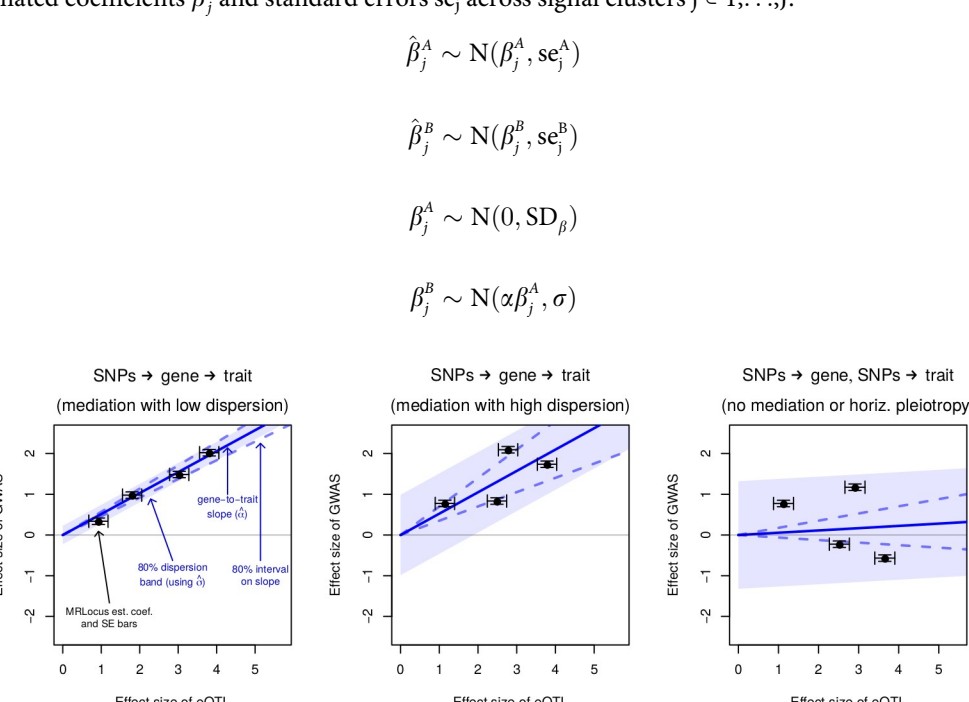

**Fig 1.** MRLocus estimates the gene-to-trait effect (solid blue line) as the slope from paired eQTL and GWAS effect sizes from independent signal clusters (black points with standard error bars), here on simulated coefficients. The dispersion of allelic effects around the main gene-to-trait effect (light blue band) is also estimated. An 80% credible interval on the slope is indicated with dashed blue lines sloping above and below the solid blue line. Panels represent loci demonstrating (A) mediation with low dispersion, (B) mediation with high dispersion, and (C) colocalization of eQTL and GWAS signals but no evidence of mediation (slope credible interval overlaps 0). Investigators may wish to prioritize loci for experimental follow-up in which a typical "dosage" pattern is observed, such that alleles contributing small amounts to expression of a gene contribute small amounts to GWAS trait, and similarly for large effect alleles.

Details regarding setting of priors for the hyperparameters $SD_\beta$ (the prior for the instrument effect sizes), $\alpha$ (the gene-to-trait slope), and $\sigma$ (the dispersion of signal cluster effect sizes around the fitted line) are provided in the S1 Methods. If the signal cluster does not provide evidence of colocalization, the estimate of the GWAS coefficient from the previous step will be near 0, and this will bring the estimated slope toward 0 as well. If the eQTL and GWAS estimated coefficients are based on standardized expression and traits, then $\alpha^2$ can be interpreted as the mediated trait variance explained in the samples. Quantile-based credible intervals on the slope coefficient provide information regarding the uncertainty of the gene mediating the trait measured in the GWAS. Finally, whereas PTWAS makes use of an $I^2$ statistic [39] for quantifying the heterogeneity of the allelic effects at the locus, MRLocus estimates the dispersion ($\sigma$) of the different allelic effects around the predicted values given by the slope. Therefore, MRLocus may have high certainty on the slope (narrow credible interval for $\alpha$ not overlapping 0), while nevertheless estimating that the dispersion of allelic effects around the slope is large ($\sigma$). The tradeoff between uncertainty on the estimate of $\alpha$ and the dispersion $\sigma$ of allelic effects around fitted line naturally depends upon the number of independent signal clusters at the locus. For loci with no allelic heterogeneity, we do not recommend running MRLocus (the software will produce a warning), and do not evaluate MRLocus on such loci here.

### Choice of methods for comparison

We chose to focus on TWMR [22] and PTWAS [25] in our comparisons, as these two methods had a focus on estimation of the gene-to-trait effect, were able to run on eQTL data for a single gene and a single tissue, and required only summary statistics and LD matrices. PTWAS adds to the analysis of gene-to-trait effects an upstream fine-mapping of cis-eQTL using DAP [33], and estimation of gene-to-trait effect heterogeneity in the case of multiple independent eQTL signal clusters, employing an $I^2$ statistic that ranges between 0 and 1. The $I^2$ statistic represents, in the gene-to-trait meditation case, the percent of variance in estimated effect sizes across signal clusters that arises from true effect heterogeneity.

We additionally compared MRLocus to LDA-MR-Egger [23] and PMR-Summary-Egger [24] on the first simulation setting. Other methods that likewise determine if one or more genes may mediate traits include SMR [40,41], CaMMEL [21], MESC [1], MR-Robin [26]. SMR uses the top cis-eQTL per gene to compute gene-to-trait effects, and was extended to SMR-multi [41] to perform null hypothesis testing across multiple SNPs per gene, but it does not offer an integrated estimate of the causal effect. We were not able to run CaMMEL using only LD matrices from the eQTL and/or GWAS cohort, as the fit.med.zqtl function takes genotype design matrices as input. We were not able to run MESC with less than 5 genes, while our focus with MRLocus is on single gene mediating effect estimation. MR-Robin also provides robust estimates of gene-to-trait effects but with a focus on multiple-tissue eQTL summary statistics as input.

### Simulation

For simulation, we used the pre-existing TWAS simulation framework, twas_sim [42], which simulates eQTL and GWAS datasets using real genotype data (1KG EUR Phase3) and outputs summary statistics. This software has a number of options for simulation parameters including *cis* heritability of gene expression (referred to here as "$h^2g$"), expression mediated trait heritability (referred to here as "$h^2med$"), and the number of SNPs in a locus which are *cis* eQTL. A single gene's expression was simulated per experiment, and both gene and trait were scaled to unit variance for slope estimation. The key twas_sim simulation equations follow. For a study with N individuals, concerning a locus with n SNPs, let the distribution of eQTL effect sizes

for causal SNPs j ∈ C, |C| = $n_{causal}$ < n follow a Normal distribution with mean and variance given by:

$$\beta_j^{eQTL} \sim N(0, h^2 g / n_{causal}).$$

The generation of a vector of gene expression $y_{gene}$ for N individuals is given by:

$$s_{gene}^2 = \hat{V}(Z \beta^{eQTL})$$

$$s_{err}^2 = s_{gene}^2 ((1/h^2 g) - 1)$$

$$\varepsilon_{indiv} \sim N(0, s_{err}^2)$$

$$y_{gene} = Z \beta^{eQTL} + \varepsilon,$$

where $\hat{V}$ is a function for the unbiased sample variance, Z is an N x n matrix of zero-centered, unit variance genotypes, $\beta^{eQTL}$ is a column vector of sampled true eQTL effect sizes which is equal to 0 everywhere except for the causal SNPs $\beta_j^{eQTL}$, and ε is a column vector of errors for individuals 1 to N. A column vector of errors, ε, are drawn from a zero-centered normal with variance $s_{err}^2$. Finally, $y_{gene}$ is standardized to unit variance before eQTL coefficients are estimated. The simulated GWAS trait $y_{trait}$ for a separate set of individuals is generated as above but using $h^2 med$ in place of $h^2 g$, and where the same set of causal (non-zero effect size) SNPs are used for $\beta^{eQTL}$ and $\beta^{GWAS}$:

$$s_{trait}^2 = \hat{V}(Z \beta^{GWAS})$$

$$s_{err,trait}^2 = s_{trait}^2 ((1/h^2 med) - 1)$$

$$\varepsilon_{indiv,trait} \sim N(0, s_{err,trait}^2)$$

$$y_{trait} = Z \beta^{GWAS} + \varepsilon_{trait}.$$

This provides gene expression and trait with population-level heritability given by $h^2 g$ and $h^2 med$ (the simulated individuals are drawn from an infinite population and so the heritability in the finite sample may differ). Finally, the true gene-to-trait slope α is calculated as the ratio of $\beta_j^{GWAS}$ / $\beta_j^{eQTL}$ for causal SNPs j, after scaling coefficients by the sample standard deviation for $y_{gene}$ and $y_{trait}$. Therefore, in the simulations we have the approximate relationship (see also S1 Method):

$$h^2 med \cong h^2 g \; \alpha^2$$

Simulations of paired eQTL and GWAS datasets were performed where the gene was a partial mediator of the trait, as well as null simulations where the gene expression was unrelated to the downstream trait and the locus contains multiple trait-only association signals (the same causal SNP percent as for eQTL). An additional simulation was performed in which the gene is a partial mediator of the trait, and three large effect trait-only association signals are added to the causal SNP set for simulating the GWAS study. This simulation was performed to assess the methods' accuracy in the presence of horizontal pleiotropy, and code for this simulation can be found in the 'hp' branch of the forked repository: https://github.com/mikelove/twas_sim/tree/hp. Null simulations were obtained by running twas_sim with two different

seeds on the same locus to produce a set of GWAS trait signals not mediated by gene expression: a distinct set of causal SNPs with unrelated effect sizes.

TWMR, PTWAS, MRLocus, eCAVIAR-MRLocus, LDA-MR-Egger, and PMR-Summary-Egger were all run on the same summary data from eQTL and GWAS simulations. The Snakemake software [43] was used for automation of simulation scripts including specification of random seed for each of the simulations, in order to assist with computational reproducibility of simulations. Simulation and analysis code is provided at https://github.com/mikelove/mrlocusPaper. The sample sizes for eQTL and GWAS were kept at their default values of $N_{eQTL}$ = 500 and $N_{GWAS}$ = 100,000, respectively, and an additional simulation was performed with $N_{eQTL}$ = 1,000. The percent of SNPs in a locus which are eQTLs was set to 1%. The parameter $h^2g$ was varied from its default value (0.1) to a higher value (0.2) and a lower value (0.05), and the expression mediated trait heritability ($h^2med$) was varied from its default value (0.01) to two lower values (0.005, 0.001, as well as to 0 indicating a null simulation where gene expression does not explain variation in a GWAS trait). Each combination of 3 (for $h^2g$) x 4 (for $h^2med$) resulted in 12 simulations (with labels A-I, Null-0.1, Null-0.2, and Null-0.05) plus two additional simulations for partial mediation with horizontal pleiotropy (HP) and higher $N_{eQTL}$ (High-N), totaling 14 simulation types (Fig A in S1 Text). 20 iterations of each simulation were drawn, and if this did not produce at least 18 iterations with allelic heterogeneity (two or more index eSNPs with p-value < 0.001 and pairwise $r^2$ < 0.05), then 20 more iterations were drawn. 40 iterations of the main (A) simulation, the horizontal pleiotropy (HP) simulation, and the higher $N_{eQTL}$ (High-N) simulation were drawn. If a method did not produce output for an iteration due to insufficient input data, e.g. no signal clusters with PIP > 0.5 for PTWAS, or < 2 nearly-LD-independent signal clusters for MRLocus, then a slope estimate of 0 is plotted in the simulation accuracy figures. The error rate and coverage for a method was calculated only over those iterations that the method produced an estimate. All methods except for PMR-Summary-Egger ran without error on all iterations.

LD-based clumping implemented in PLINK (v1.90b) [32] was performed on eQTL simulation un-adjusted p-values with the following settings:—clump-p1 0.001—clump-p2 1—clump-r2 0.1—clump-kb 500. All PLINK clumps were provided to TWMR (commit 62994ec) [22] and MRLocus (v0.0.22) for gene-to-trait effect estimation. PTWAS (v1.0) [25] was provided with output from DAP (DAP-G, commit ac38301) [33] with settings: -d_n $N_{eQTL}$ -d_syy $N_{eQTL}$ (as twas_sim scales expression to unit variance). PTWAS code was modified (at line 97 for commit b5714f3) to allow for input of estimated coefficients and their SE, such that it provided slope estimates on the original scale of coefficients, not $z$-scores. LDA-MR-Egger and PMR-Summary-Egger (v1.0) were supplied with the eQTL and GWAS summary statistics of the locus and the LD matrix, without clumping. For 1–2 simulation iterations, LDA-MR-Egger or PMR-Summary-Egger would output a very large estimate of $\alpha$ (> 2 in absolute value), and these large estimates were removed to give more representative error rates for these two methods. PMR-Summary-Egger does not provide an SE of the causal effect in its output so was excluded from coverage evaluation. eCAVIAR (v2.2) was run per nearly-LD-independent signal cluster with -c 1 (maximum of one causal SNP), without collapsing of highly correlated SNPs, as an alternative to MRLocus colocalization step. For eCAVIAR-MRLocus, the SNP with highest CLPP among those with p-value less than the PLINK clumping threshold was provided from each signal cluster to the slope fitting step. For all simulations, if there were no SNPs in the simulated locus with eQTL un-adjusted p-value < 0.001 then a new seed was drawn. Furthermore, all methods were only evaluated on loci with evidence of allelic heterogeneity, as demonstrated by more than one nearly-LD-independent ($r^2$ < 0.05) signal cluster with index eQTL p-value < 0.001. For simulation comparisons, MR with inverse variance weighted (IVW) regression with fixed effects [44] was computed using the true causal eQTLs

("causal") or across all SNPs ("all") using the mr_ivw_fe function in the TwoSampleMR R package (v0.5.5) [45] with eQTL as the exposure study and GWAS as the outcome study. The "causal" IVW MR analysis served as an "oracle" estimator in the simulations, as it was provided with information not available in a typical analysis and not provided to other methods. The number of true eQTL SNPs, PLINK clumps, DAP signal clusters, and clusters passing 1st and 2nd round of $r^2$-based trimming are shown in Fig B in S1 Text. The distribution of the number of SNPs per clump before and after collapsing highly correlated SNPs (part of MRLocus preprocessing, described in S1 Methods) is shown in Fig C in S1 Text.

In order to assess whether trimming signal clusters to obtain pairwise $r^2 < 0.05$ would be sufficient to protect against false positives for MRLocus slope fitting resulting from correlated instruments, a simulation was performed across a range of $r^2$. Estimated eQTL and GWAS coefficients for between J = 4, 6, or 8 signal clusters were generated from a multivariate normal distribution, such that the population correlation between adjacent signal clusters took on a specific $r^2$ value, and all other pairs had correlation of 0. The mean vector for the eQTL (A) study was centered on [10,. . .,10], while the mean vector for the GWAS (B) study was centered on the origin, such that the true gene-to-trait slope is α = 0. These coefficients and their standard errors were provided to the slope estimation step of MRLocus, and 80% credible intervals were calculated for 400 iterations per combination of J and $r^2$. The code for this simulation is provided in the corr_instr_sim directory of the mrlocusPaper GitHub repository.

By producing signal clusters with PLINK clump using—clump-p1 0.001, this ensures that the index eSNP for each signal cluster has an absolute Z-score of > ~3.3, or equivalently an F-statistic > ~10.8. In order to assess whether even stronger criteria on the strength of the signal clusters provided to MRLocus would improve accuracy of the gene-to-trait effect, we re-ran PLINK clump on the main (A) simulation and the high $N_{eQTL}$ simulation using—clump-p1 0.0001 (F-statistic > ~15.1), and re-ran TWMR and MRLocus. Such estimates for TWMR and MRLocus are referred to using "_p1e-4" in the Results. As PTWAS did not always produce estimates on simulated loci due to insufficient signal clusters with posterior inclusion probability (PIP) > 0.5, we additionally tested PTWAS on the A and high $N_{eQTL}$ simulations using signal clusters with PIP > 0.1, referred to as "_t0.1" in the Results.

Two additional assessments of the simulation datasets were performed, to better understand underlying factors that may explain differences in method performance. For MRLocus and eCAVIAR colocalization steps, the accuracy in recovering the causal eSNPs was assessed across the different simulation settings. Specifically, for signal clusters containing a true causal eSNP, the SNP chosen as the candidate instrument from each signal cluster for slope fitting was evaluated as to whether it was equal to or in high correlation (r > 0.95) with the causal eSNP. The accuracy was then computed by averaging over all signal clusters from all iterations of the simulation. Additionally, the coverage of true values by confidence or credible intervals was assessed using information about the true simulated value. As insufficient coverage could occur due to estimation bias or badly calibrated standard errors, for three simulations with $h^2g$ of 10% (A, D, G), we calculated the bias for TWMR, PTWAS, MRLocus, and eCAVIAR-MRLocus, using the sample mean of the estimates and the population-level true alpha value of $(h^2med / h^2g)^{-1/2}$. We then re-computed the interval coverage after subtracting the bias. While these "oracle" bias-adjusted coverage values could not be obtained in real settings, they help to answer whether correction of estimation bias alone would resolve problems with insufficient coverage.

## Real data analysis

To evaluate the performance of methods on real data, we applied TWMR, PTWAS, and MRLocus to five candidate causal genes for mediation of particular GWAS traits, chosen based

on literature review of previous connections between gene expression and downstream phenotype: *SORT1* (liver)—low-density lipoproteins (LDL), *MRAS* (tibial artery)—coronary artery disease (CAD), *PHACTR1* (tibial artery)—CAD, *CETP* (liver)—high-density lipoproteins (HDL), and *LIPC* (liver, as well as in blood)—HDL. eQTL and GWAS summary data were obtained as summarized in Table A in S1 Text. Briefly, eQTL data on tibial artery, blood, and liver was obtained from the Genotype-Tissue Expression (GTEx) project v8 [46], eQTLGen Consortium 2019-12-23 release [47], and directly from the authors of a liver meta-analysis study [48], respectively. GTEx v8 estimated eQTL coefficients were on the scale of unit variance expression values (following per-gene inverse normal transformation), eQTLGen estimated coefficients and standard errors were derived from z-scores and MAF as indicated by the README for SMR input, while the liver meta-analysis estimated coefficients were on the scale of $\log_2$ normalized expression, according to references. GWAS summary statistics on CAD were obtained from CARDioGRAMplusC4D (Coronary ARtery DIsease Genome wide Replication and Meta-analysis (CARDIoGRAM) plus The Coronary Artery Disease (C4D) Genetics) consortium [49], where coefficients represent log odds ratios (OR), and the UK Biobank (https://www.ukbiobank.ac.uk) obtained from http://www.nealelab.is/uk-biobank/, where coefficients are estimated with respect to unit variance, continuous scale HDL or LDL.

Prior to defining independent signal clusters, we filtered out SNPs with MAF < 0.01 from GWAS data, and p-values were corrected with lambda GC [50]. We used raw p-values for eQTL data and genomic control corrected p-values for GWAS data. As our task for downstream inference is estimation of the slope of GWAS coefficients over eQTL coefficients across signal clusters, it is not necessary that the eQTL signal clusters attain genome-wide significance before being provided to MRLocus. We first identified nearly-LD-independent signal clusters for eQTL by LD-based clumping implemented in PLINK (1.90b) [32] with the following settings:—clump-p1 0.001—clump-p2 1—clump-kb 500—clump-r2 0.1 (as in the simulation setting). LD was estimated in the European population from the 1000 Genome Project phase 3 (1KG EUR). Next we directly obtained test statistics of matched SNPs in each eQTL cluster from GWAS summary data.

### Generating MRLocus input files

For each independent pair, we generated MRLocus input files (effect size tables) with estimated coefficients, SE, and reference and effect alleles from both eQTL and GWAS. Pairwise LD (Pearson's r) between SNPs were generated by the PLINK—r function for all SNPs included in the corresponding effect size table using 1KG EUR. We also included major/minor allele information from the reference panel (PLINK bim files) in the effect size table for proper allele flipping for statistical modeling and visualization (S1 Methods).

### Real data analysis with other methods

TWMR and PTWAS were used to compare estimates on real data loci. Both software packages require definition of eQTL signal clusters and we used DAP (DAP-G, commit ac38301) to estimate independent eQTL clusters for PTWAS (v1.0) as described in the original paper, while for TWMR (commit 62994ec) we ran PLINK clumping with the same parameters we used for MRLocus (v0.0.22) (c.f. the original TWMR paper performed approximate conditional analyses). DAP was applied to estimated coefficients and SE from eQTL summary statistics with default options except we set maximum models (-msize) to 20. For the *CETP* locus, there were not any signal clusters for PTWAS with sufficient posterior inclusion of probability (all signal cluster PIP < 0.5), so the threshold was lowered to 0.1 in order to visualize an estimate and interval.

# Results

## Simulation

In order to evaluate the accuracy in estimating the mediated gene effect, we used a GWAS and eQTL simulation. The twas_sim simulation framework used for evaluating methods has default values for cis eQTL gene expression heritability of 10% ($h^2g$) and expression mediated trait heritability of 1% ($h^2med$). As the eQTL-based gene expression heritability could feasibly be higher or lower than 10%, we investigated values of 20% and 5% as well, which are within the range of detection for eQTL studies with hundreds of samples [51–55]. The default mediated trait variance explained value of 1% for a single gene is likely high, given that a recent publication using GTEx data has estimated the proportion of heritability mediated by expression across all genes to be around 11% (±2%) averaging over 42 traits, with the top mediated traits having around 30% of heritability mediated by expression [1]. We therefore considered even lower expression mediated heritability of the GWAS trait, $h^2med$ of 0.5% and 0.1%, as well as a null simulation where gene expression did not mediate the downstream trait in any way (0%). We note that the selection of simulated loci with two or more nearly-LD-independent eQTL with index eSNP p-value < 0.001 may result in Winner's Curse for low-powered studies, where true values tend to be smaller than the estimated coefficients for the eSNPs passing a significance threshold. Over-estimation of instrument effect sizes would then result in under-estimation of the mediation effect. There was some over-estimation for the lowest gene heritability ($h^2g = 0.05$), but a balance of both under- and overestimation of effect sizes for $h^2g$ of 0.1 or 0.2 (Fig D in S1 Text).

For the default values of 10% gene $h^2$ and 1% $h^2med$ (simulation A), eCAVIAR-MRLocus had the highest accuracy in terms of relative mean absolute error (RMAE), dividing the error by the absolute value of the true slope, followed by MRLocus and PTWAS (Fig 2A). TWMR and PTWAS demonstrated the most negative bias, calculated using the sample mean of the estimator and the population-level α. The bias for TWMR, PTWAS, MRLocus, and eCAVIAR-MRLocus was -0.112, -0.079, -0.003, and -0.053, respectively. In addition, only eCAVIAR-MRLocus and MRLocus were able to achieve nominal coverage for 80% credible

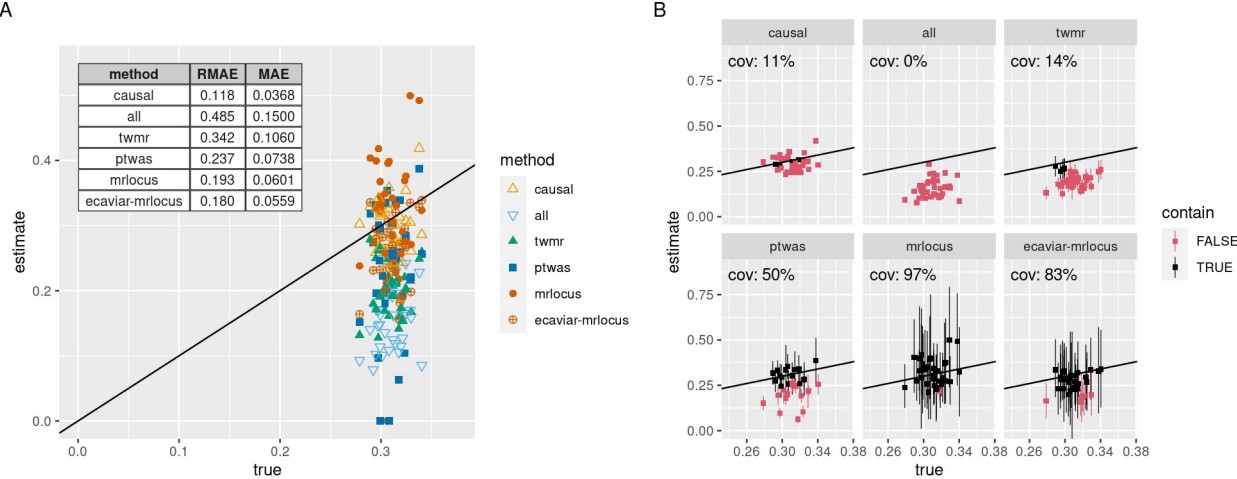

**Fig 2. Performance of methods on simulated eQTL and GWAS datasets.** (A) Estimates of each method over true (simulated) gene-to-trait values. The method denoted with "causal" indicates an inverse variance weighted slope estimation using the true causal SNPs but the estimated coefficients and SEs (an oracle estimate), and "all" indicates an inverse variance weighted slope estimation using all SNPs. (B) Observed coverage (abbreviated "cov.") of 80% confidence or credible intervals from each method. If the interval contains the true effect size, it is colored black, otherwise red.

intervals over the true values, while other methods had lower observed coverage, having too narrow confidence intervals (Fig 2B). We additionally tested two other methods, LDA-MR-Egger and PMR-Summary-Egger at the default twas_sim settings (Fig E in S1 Text). These additional two methods had higher error compared to other methods at the default twas_sim settings (with LDA-MR-Egger having lower error of the two), and so we focused on the latter three methods for further evaluation. Here, and in all simulation settings considered, the evaluated methods had higher RMAE of gene-to-trait slope compared to an oracle method that uses only the true causal SNPs (which are generally not known) and their estimated coefficients ("causal") in IVW regression with fixed effects [44].

In the evaluation of pairwise $r^2$ across signal clusters, for loci with 4, 6, or 8 signal clusters, increase in the $r^2$ of adjacent signals was associated with an increase in the false positive rate for MRLocus (Fig F in S1 Text). False positive rate was measured by the rate of 80% credible intervals not covering the true value of $\alpha = 0$, for 400 simulation iterations per combination of simulation parameters. At $r^2 < 0.05$, the average rate of intervals not covering 0 was approximately 20%, so MRLocus was then achieving the nominal level. We therefore recommend trimming signal clusters to achieve $r^2 < 0.05$ across pairs of clusters before running MRLocus slope estimation, because allowing higher pairwise $r^2$ across instruments could lead to loss of nominal interval coverage. A convenience function in the MRLocus package, trimClusters, can be used to prioritize signal clusters with higher strength of eQTL signal. Violin plots of the pairwise $r^2$ across clusters, for all instruments passed to MRLocus slope fitting is provided in Fig G in S1 Text.

We modified simulation A in two ways, to assess how horizontal pleiotropy or higher sample size would worsen or improve estimation of partial mediation, respectively. The effect of horizontal pleiotropy, i.e. signals associated separately with gene expression and trait, on estimation of gene-to-trait mediation was assessed by adding non-expression associated GWAS signals to a loci exhibiting partial mediation. An example region plot for one of the HP simulation iterations is provided in Fig H in S1 Text. TWMR, PTWAS, MRLocus, and eCAVIAR-MRLocus had higher error and lower interval coverage for the horizontal pleiotropy simulation compared to their performance in simulation A, as expected (Fig I in S1 Text, with additional methods provided in Fig J in S1 Text). MRLocus and eCAVIAR-MRLocus had nearly the same RMAE, which was the lowest among the methods tested. LDA-MR-Egger was less affected by inclusion of trait-only signals, although still having higher error than all other methods except for PMR-Summary-Egger. MRLocus and eCAVIAR-MRLocus obtained close to 80% coverage (86% and 76%, respectively) despite the addition of coincident trait-only GWAS signals. Finally, for the same default $h^2g$ and $h^2med$ settings as simulation A, we increased the sample size to $N_{eQTL} = 1,000$ to see how higher power in detection of instruments may translate to better estimation of mediation. In this higher power setting, all methods had lower error in estimating the gene-to-trait effect (Fig K in S1 Text). Here, MRLocus had the lowest RMAE, slightly below that of eCAVIAR-MRLocus, and both obtained nominal interval coverage. PTWAS had improved confidence interval coverage (from 28% to 50%) while other methods had similar coverage as with $N_{eQTL} = 500$.

For simulations using other parameter settings of gene expression heritability $h^2g$ and expression mediated trait heritability $h^2med$, MRLocus had the second lowest RMAE in 6 of the remaining 8 non-null simulation settings with non-default values of $h^2g$ and $h^2med$ (Figs L-S in S1 Text). In simulation G, with 10% $h^2g$ and 0.1% $h^2med$, PTWAS had lower RMAE than MRLocus, and in simulation H, with 20% $h^2g$ and 0.1% $h^2med$, eCAVIAR-MRLocus had lower RMAE than MRLocus and PTWAS. Overall, considering the non-null simulations performed, MRLocus or eCAVIAR-MRLocus had the lowest RMAE in estimating the gene-to-trait effect compared to other methods for 10 out of 11 of the settings (Fig 2, and Figs I, K-P,

R-S in S1 Text). When expression heritability and mediated heritability were moderate to high ($h^2 g \geq 10\%$, $h^2 med \geq 0.5\%$), these two methods would come close to the RMAE of the oracle estimate ("causal").

In terms of credible or confidence interval coverage, MRLocus and eCAVIAR-MRLocus always had better or equal coverage of the true values compared to all other methods within the non-null simulation settings, though it did not always reach the nominal level (Figs T-AA in S1 Text). When expression heritability was low ($h^2 g = 5\%$), or the mediated trait heritability low ($h^2 med = 0.1\%$) MRLocus and eCAVIAR-MRLocus had coverage roughly in the range 50–70%; exceptions to this trend were that MRLocus achieved >80% coverage for simulation F, and both methods had >80% coverage for simulation H. MRLocus always had higher coverage than eCAVIAR-MRLocus when they differed by more than 5%. PTWAS tended to have higher coverage than TWMR, but they each had a maximal coverage of 39% and 26%, respectively; TWMR and PTWAS had the best coverage on the $h^2 g = 20\%$, $h^2 med = 0.1\%$ simulation. The oracle method tended to have narrow confidence intervals, and MRLocus had higher coverage than the oracle method for all simulations. The maximal coverage of the oracle was simulation F with 40%. We believe the lower-than-nominal coverage seen here for the oracle method is likely from insufficient propagation of error during slope estimation. As the twas_-sim framework does not include heterogeneity of effects from different signal clusters, an additional simulation was performed to assess MRLocus's estimation of the dispersion of effects around the gene-to-trait fitted line (Fig AB in S1 Text). Here, MRLocus was accurate both in estimation of the dispersion and quantification of uncertainty (attaining nominal credible interval coverage), with higher accuracy and smaller intervals as the number of nearly-LD-independent clusters increased, as expected.

In the 3 null simulation settings with $h^2 g$ of 5%, 10%, 20% but no mediated heritability, and trait-only GWAS signals, MRLocus always had the highest interval coverage of the true slope value of 0 (average of 79%, close to the nominal 80%), followed by eCAVIAR-MRLocus (average of 63%) (Fig AC in S1 Text). Here, TWMR had average coverage of 35% and PTWAS had average coverage of 38%. The oracle method with the true eQTL SNPs had average coverage of 14%. The oracle method again had too narrow intervals as in the non-null simulations. TWMR and PTWAS had lower mean absolute error (MAE) for estimating $\alpha = 0$ than MRLocus and eCAVIAR-MRLocus, which was expected due to the bias toward 0 seen in the non-null simulations.

We performed sensitivity analysis for the two thresholds involved in the formation of signal clusters: the eQTL p-value used with PLINK clump (default of p-value < 0.001), and the PIP threshold for inclusion of a signal cluster to PTWAS causal effect estimation (default of PIP > 0.5). While p-value < 0.001 corresponds to an F-statistic > ~10.8, use of stronger criteria for the eQTL instruments (p < 0.0001, F-statistic > 15.1) did not result in higher accuracy or better interval coverage for MRLocus in simulation A or with higher $N_{eQTL} = 1,000$ (Fig AD in S1 Text). However, in both simulation settings, accuracy and coverage were improved slightly for TWMR using the stricter p-value for clumping. Running PTWAS with PIP threshold lowered from the default value of 0.5 to 0.1 did not improve accuracy or interval coverage in these simulation settings.

We assessed whether MRLocus colocalization step or eCAVIAR was more accurate at identifying the causal eSNP in non-null and null simulations. eCAVIAR outperformed MRLocus on the non-null simulations in its ability to detect the true causal SNP, for all 11 non-null simulations (Fig AE in S1 Text). We note that eCAVIAR does not provide posterior estimates of eQTL and GWAS effect sizes, the originally designed input to the MRLocus slope fitting step. When providing eCAVIAR-chosen SNPs and the original estimated coefficients from eQTL and GWAS, we find that eCAVIAR-MRLocus is more accurate for only 2 out of 11 non-null

simulations (simulation A and H). Furthermore, MRLocus generally had higher coverage of the gene-to-trait effect compared to eCAVIAR-MRLocus in the more difficult simulation settings (simulations C, F, and I, while eCAVIAR-MRLocus had higher coverage for simulation G). For the null simulations, in which eQTL eSNPs and GWAS trait-associated SNPs are distinct and expression does not mediate trait variance, MRLocus colocalization was always more accurate at identifying the true causal eSNP, and the MRLocus-chosen eSNPs were better for providing intervals that cover the true gene-to-trait effect ($\alpha = 0$).

Insufficient confidence or credible interval coverage could result from estimator bias, and TWMR and PTWAS exhibited downward bias in the simulations. We further examined if calculating and removing the estimator bias (using the true, population-level slope $\alpha$) improved interval coverage for TWMR, PTWAS, MRLocus and eCAVIAR-MRLocus, for the simulations with $h^2_g$ of 10% (simulation A, D, and G). In some cases, bias removal helped substantially, for example TWMR's coverage rising from 14% to 67% in simulation A and from 8% to 59% in simulation D, or PTWAS's coverage rising from 35% to 76% in simulation G (Fig AF in S1 Text). Still, even with this "oracle" bias removal, TWMR and PTWAS still had cases of insufficient coverage, for example PTWAS with 32% coverage in simulation D, and TWMR with 44% coverage in simulation G. This indicated that the insufficient coverage observed for these methods was not only due to bias, but also from too small standard errors on the causal effect estimate.

We assessed method timing across the 80 main (A) and high $N_{eQTL}$ simulations. MRLocus was approximately 17 times slower than TWMR and 32 times slower than PTWAS, with a mean running time per locus of 75 seconds compared to 4.3 seconds for TWMR and 2.3 seconds for PTWAS (Fig AG in S1 Text). The additional elapsed time is because MRLocus involves a colocalization step with model fitting per signal cluster using MCMC. MRLocus colocalization step was provided with four cores for running four MCMC chains, while other methods and MRLocus slope fitting used a single core. Performing colocalization with eCAVIAR reduced the running time by about two-thirds for eCAVIAR-MRLocus compared to MRLocus (3.4 seconds for eCAVIAR colocalization and 22 seconds for MRLocus slope fitting, so in total 6 times slower than TWMR and 11 times slower than PTWAS). One reason why MRLocus colocalization is slower than eCAVIAR is that MRLocus's model for colocalization adapts its priors for eQTL and GWAS effect sizes to the data in each signal cluster in terms of the scale and sparsity of causal variants. Runtime for PLINK clumping and DAP were not included in the times presented above.

## Real data analysis

We compared TWMR, PTWAS, MRLocus, and eCAVIAR-MRLocus using eQTL and GWAS summary statistics for five gene-trait pairs in which there is strong evidence that the gene mediates the trait, and the direction of the effect has also been reported, such that we could compare our estimates against the literature. For one gene-trait pair (*LIPC* and HDL), we examined the effect in liver, the tissue reported in literature as the relevant gene expression context, as well as using eQTL in blood, where the expression association signal may or may not provide strong evidence of mediation of the GWAS trait. LocusZoom-style plots [56,57] for the eQTL and GWAS tracks are provided in Figs AH-AL in S1 Text. Additionally, the LD patterns for all trimmed signal clusters are provided in Figs AM-AR in S1 Text.

On the six eQTL-GWAS dataset pairs, all four methods had consistent sign of the mediating effect (Fig 3). MRLocus and eCAVIAR-MRLocus generally had larger credible intervals compared to confidence intervals from the other two methods, as was seen in the simulation datasets where MRLocus and eCAVIAR-MRLocus had improved coverage of the true effect size.

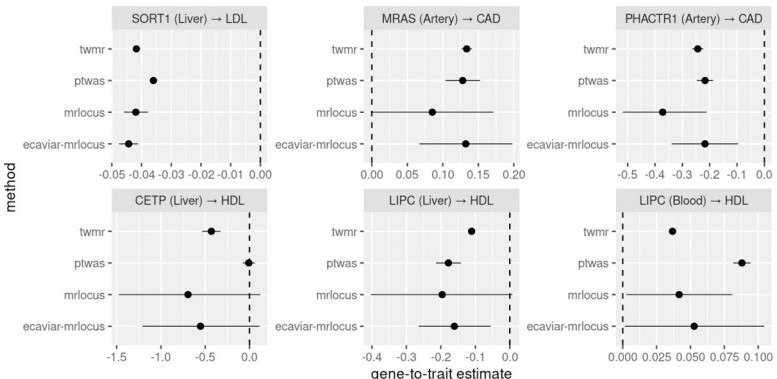

**Fig 3. Estimated gene-to-trait effects for TWMR, PTWAS, and MRLocus on five eQTL and GWAS datasets for strong candidate genes for mediation of the GWAS trait and one auxiliary tissue example (*LIPC* in blood).** 80% confidence or credible intervals are shown (MRLocus provides quantile-based credible intervals). The artery eQTL were estimated from inverse normal transformed expression data from GTEx, transformed z-scores for eQTLGen, while the liver meta-analysis eQTL were estimated from log2 expression data, thus the estimated slopes represent changes in SD of artery or blood gene expression on log odds for CAD risk and on SDs of lipid levels, and doubling of liver gene expression on SDs of lipid levels.

In all cases except *LIPC* in blood, the sign of the mediating effect was in concordance with literature: higher *SORT1* (liver) expression decreasing LDL levels [19,58,59], higher *MRAS* (artery) being hazardous for CAD [60–62], higher *PHACTR1* (artery) being protective for CAD [63,64], higher *CETP* (liver) decreasing HDL levels [65], and higher *LIPC* (liver) decreasing HDL levels [66]. For *LIPC* assayed in blood, TWMR and PTWAS have positive effect sizes with narrow confidence intervals (opposite of the mediating effect for *LIPC* in liver), while MRLocus and eCAVIAR-MRLocus have large credible intervals, reflecting the conflicting evidence across blood eQTL signal clusters (Figs AS and AT in S1 Text). We note that MRLocus also has large credible intervals for *LIPC* in liver, due to a limited number of nearly-LD-independent signal clusters (n = 3) and posterior GWAS effect sizes close to 0 for two signal clusters (Fig AS in S1 Text). For *CETP* (liver) paired with HDL, PTWAS did not have any DAP signal clusters with sufficient PIP (threshold of 0.5) to estimate an effect, so the threshold was lowered to 0.1, which produced an estimate close to 0, while MRLocus and eCAVIAR-MRLocus likewise displayed high uncertainty of mediation given the limited number of signal clusters (n = 2 for both, Figs AS and AT in S1 Text).

MRLocus had the strongest evidence for mediation with *SORT1* (liver) and its effect on LDL, with LocusZoom-style plot of the region in Fig 4A and MRLocus gene-to-trait estimate plot in Fig 4B. The number of signal clusters used for MRLocus for *SORT1* (liver), *MRAS* (artery), *PHACTR1* (artery), *CETP* (liver), *LIPC* (liver), and *LIPC* (blood) were: {6, 9, 5, 2, 3, 31}, and for eCAVIAR-MRLocus were: {5, 5, 6, 2, 3, 28}. The number of nearly-LD-independent signal clusters for blood eQTL for *LIPC* (n = 31 and 28 for MRLocus and eCAVIAR-MRLocus, respectively) is likely an overestimate of the number of true causal SNPs at this locus, and new methods for clumping or summary-based conditional analysis large-sample-size meta-analysis eQTL studies may refine these signals. MRLocus and eCAVIAR-MRLocus also had strong evidence for mediation with *PHACTR1* (artery) and its effect on CAD. Two loci where MRLocus and eCAVIAR-MRLocus have slightly differing results were *MRAS* (artery) on CAD and *LIPC* (liver) on HDL, where MRLocus-selected instruments tended to have more scatter (Fig AS in S1 Text), and thus more uncertainty reflected in the credible interval of the gene-to-trait effect.

Estimates of heterogeneity and dispersion for PTWAS and MRLocus are shown in Table 1. PTWAS indicates heterogeneity of instruments using an $I^2$ statistic that ranges from 0 to 1

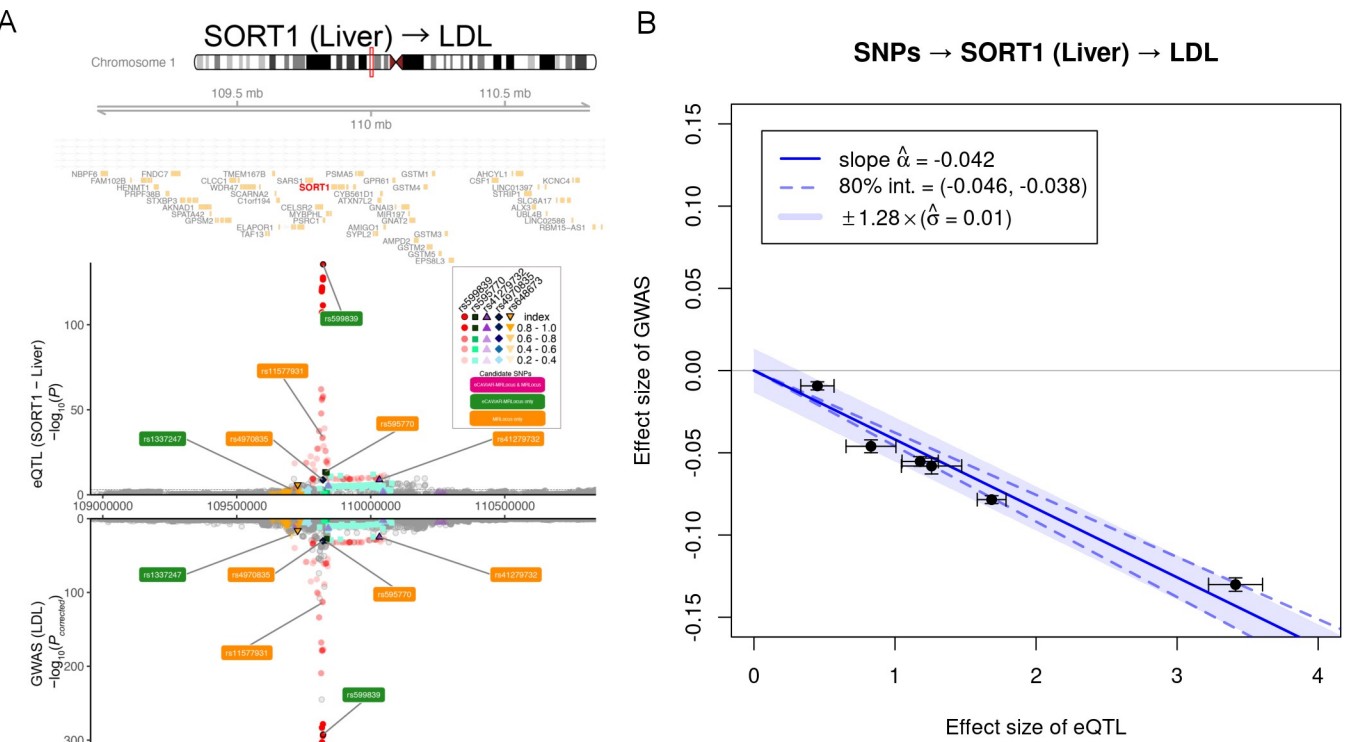

**Fig 4. Colocalization and MRLocus estimation for *SORT1*.** (A) Colocalized signals in the *SORT1* region. From top panel to bottom, gene model (NCBI Refseq), eQTL for *SORT1* in liver (N = 588) (Strunz et al., 2018) and LDL association within UKBB (N = 343,621). Dashed line indicates a significance threshold at p = 0.001 or p = 5x10$^{-8}$ for eQTL and GWAS respectively. Colored labels indicate eSNPs used for slope fitting with both methods, eCAVIAR-MRLocus, or MRLocus. (B) MRLocus plot of the gene-to-trait effect for *SORT1* expression in liver on LDL levels. The signal clusters all provide consistent evidence for a gene-to-trait effect of -0.042, meaning that doubling of gene expression level in liver should reduce LDL by 4.2% of its population standard deviation. An 80% credible interval for the slope is indicated by dashed blue lines around the solid blue slope, while a range of heterogeneity of allelic effects is indicated by the light blue band.

(low to high heterogeneity). MRLocus estimates the dispersion of instruments around the fitted line on the scale of the mediation effect; the estimate can be compared across loci by calculating the mean mediated effect: (estimate α) x (mean of estimated coefficients for $\beta_j^A$). Visually, the mean mediated effect is the y-axis value from the fitted slope in Fig 4B corresponding to the middle values of estimated eQTL coefficients. The ratio of the estimate of dispersion σ over the absolute value of mean mediated effect indicates relatively how much the effects from nearly-LD-independent signal clusters vary around the conditional mean. The lowest I$^2$ for PTWAS was for *MRAS* and *LIPC* (liver), which each had an estimate of 0 for

**Table 1. I$^2$ statistic for PTWAS and dispersion estimate σ from MRLocus for eQTL-GWAS pairs.**

| Gene (tissue)—trait | PTWAS I$^2$ | MRLocus σ | Mean mediated | σ / \|MM\| |
|---|---|---|---|---|
| *SORT1* (liver)—LDL | 0.85 | 1.47 | -0.062 | 0.17 |
| *MRAS* (art.)—CAD | 0.00 | 0.16 | 0.014 | 0.91 |
| *PHACTR1* (art.)—CAD | 0.76 | 0.12 | -0.044 | 0.71 |
| *CETP* (liver)—HDL | 0.92 | 0.17 | -0.12 | 1.43 |
| *LIPC* (liver)—HDL | 0.00 | 0.14 | -0.028 | 1.30 |
| *LIPC* (blood)—HDL | 0.29 | 0.12 | 0.0051 | 7.16 |

Additional columns provide the mean mediated effect for MRLocus, and the ratio of the dispersion to the absolute value of the mean mediated effect.

heterogeneity. The lowest dispersion relative to the mean mediated effect for *MRLocus* was for *SORT1*, followed by *PHACTR1* and *MRAS*.

On the six gene-tissue-trait combinations, eCAVIAR colocalization took an average of 6 seconds per locus, MRLocus slope fitting following eCAVIAR took an average of 18 seconds per locus, and MRLocus (both steps) took an average of ~4 minutes per locus, using 4 cores for MRLocus colocalization step.

## Discussion

Here, we introduce MRLocus, a two-step Bayesian statistical procedure for estimation of gene-to-trait effects from eQTL and GWAS summary statistics. We find that MRLocus tends to have high accuracy in estimating the gene-to-trait effect across a variety of simulation settings, often higher than existing methods, and always had better credible interval coverage of true values, whether in partial mediation or null simulations. On real data analyses, MRLocus and eCAVIAR-MRLocus (using eCAVIAR for colocalization followed by MRLocus slope fitting) had consistent sign of estimates and comparable effect size compared to TWMR and PTWAS, but larger credible intervals compared to the other methods' confidence intervals. While the effect sizes of alleles detected by GWAS on downstream traits examined here may be moderate, MRLocus's estimation of the causal effect from perturbation of gene expression can be helpful in assessing the impact of therapeutic effects modulating expression on downstream traits [67]. For various systems, different downstream trait effect sizes qualify as practically meaningful increases or decreases, and MRLocus provides a framework for assessing what level of gene expression perturbation may be needed to obtain such changes in a downstream trait.

MRLocus was used to estimate the gene-to-trait effect following colocalization with MRLocus's own Bayesian model of eQTL and GWAS coefficients per signal cluster, and alternatively using the existing eCAVIAR method. Both approaches were found to perform well in simulations, and better than other methods in terms of accuracy and interval coverage of the true slope. Use of eCAVIAR sped up the pipeline substantially (by at least an order of magnitude after accounting for use of multiple cores), as colocalization in MRLocus is the slowest step. eCAVIAR was better on average at identifying the true causal eSNP for non-null simulations, although MRLocus was better at identifying true causal eSNP in the null simulations. In simulation settings when expression heritability was high ($h^2g > 5\%$), passing original estimated coefficients for eCAVIAR-identified SNPs to slope fitting performed just as well as passing MRLocus's posterior estimates for MRLocus-identified SNPs. However, when expression heritability was lower ($h^2g = 5\%$), MRLocus had a more substantial gain in accuracy in estimating the gene-to-trait effect over eCAVIAR-MRLocus. This advantage may be due to the shrinkage of the eQTL/GWAS effect sizes that occurs in the MRLocus colocalization step (eCAVIAR does not provide posterior estimates of effect size). Also, in the $h^2g = 5\%$ simulations, eCAVIAR-MRLocus had moderate coverage of the true effect (48–58%), while MRLocus had closer to nominal coverage (68–87%). In the real data loci, the two alternatives for colocalization input to MRLocus slope fitting had similar output, except for the case of *MRAS* (artery) and *LIPC* (liver) where MRLocus had a larger interval, intersecting with 0. Overall, we recommend MRLocus over eCAVIAR-MRLocus for general purpose estimation of gene-to-trait effects, but both are provided as options in the MRLocus software guide.

While the mediator evaluated by MRLocus in this work was gene expression effects via eQTL, the methods are generic, and protein abundance effects via pQTL could be used instead of eQTL. Two-sample MR linking pQTL and GWAS has already uncovered 30 metabolite features with evidence of causal effects on at least one disease [68], and a recent pQTL study of

hepatic proteins reported a median of 4.5 local pQTL variants per protein [69], suggesting that there are loci with sufficient number of nearly-LD-independent clusters for MR analysis. Additionally, large scale pQTL studies of similar sample size to eQTLGen (>30,000 individuals) have uncovered secondary pQTL signals upon conditional analysis for hundreds of loci [70]. Alternatively, pQTL could be used in place of the downstream GWAS trait in order to study mediation from gene expression to protein abundance [71], as previous work has found pQTL effect size to be positively correlated with eQTL effect size for variants ascertained through eQTL in human [72,73], and that colocalized eQTL and pQTL signal leads to higher observed RNA-protein correlations in mice [74].

Given MRLocus's improved performance with respect to interval coverage in the simulation, we feel that accurate estimation of uncertainty is an advantage to MRLocus, and the focus in developing a new method was on specificity for prioritization of gene targets for functional follow-up experiments. Additionally, the MRLocus model is extensible. The slope-fitting model could easily be generalized to use an alternative monotonic function, as long as there are sufficient nearly-LD-independent signal clusters to support fitting. On the other hand, MRLocus's slower speed means it is likely not the best choice for a global scan of the transcriptome for mediating genes, while the other methods examined and discussed here have been successfully used to scan across all genes and across multiple tissues. Furthermore, MRLocus is designed for investigating loci with strong causal gene candidates, whereas other methods that estimate gene-to-trait effects for many genes in a locus simultaneously may have less biased estimation of the effect, when a strong gene candidate is not present. Future work on the MRLocus model may involve estimation of the mediating effect of candidate causal genes in the context of other relevant genes in a pathway and a polygenic background [75].

There are a number of important limitations of the current study. First, the 1000 Genome Project phase 3 (1KG EUR) was used for LD calculations and clumping on the real data loci, although this reference panel is now a small sample size compared to other available resources such as gnomAD, TOPMed, and UK Biobank [76–78]. In addition, new tools such as TopLD (http://topld.genetics.unc.edu/topld/) allow for easy access to LD based on TOPMed whole genome sequencing data. MRLocus and other methods for estimating mediation effects would benefit from these larger reference panels for LD calculations. Another important limitation is that, as eQTL study sample sizes increase, the threshold of eQTL p-value < 0.001 for clumping and inclusion of eQTL instruments may prove to be too lenient, as could be seen in the large number of nearly-LD-independent loci with small eQTL effect size for *LIPC* in blood from eQTLGen (Figs AS and AT in S1 Text). More sophisticated statistical methods for performing conditional analyses from summary statistics would benefit downstream analyses such as MRLocus that make use of nearly-LD-independent signal clusters, and stricter instrument inclusion criteria may be needed for large studies and meta-analyses with ancestry heterogeneity across studies.

We note that our method focuses on common variation (MAF > 0.01), and that we collapse highly correlated SNPs to a single representative SNP during the pre-processing, such that we cannot determine if the final selected SNP is the true "causal" SNP. Future development of MRLocus could involve upstream use of methods defining credible sets [79–84] or modeling based on a posterior inclusion probability as in LLARRMA or DAP [33,85]. Therefore, the current implementation of MRLocus can perform fine-mapping to the level of a highly correlated set of SNPs, which may be sufficient for identifying one or more regulatory elements (RE) to prioritize for functional follow-up experiments. The current implementation of MRLocus assumes that the mediation slope passes through the origin, and therefore that eQTL signal clusters do not affect the downstream trait through genes other than the eGene. Estimation was nevertheless shown to be robust when loci harbored large trait-only association signals.

Further iterations of MRLocus could relax this assumption through the addition of an intercept term accounting for invalid instruments as in MR-Egger [86,87]. Complementary information linking RE to genes, e.g. as measured by Hi-C, was not examined here, but have been proven successful elsewhere (HUGIN [88], H-MAGMA [89]), and we envision that prioritization of signal clusters in MRLocus that are supported by Hi-C would increase its power to detect causal genes.

As part of the MR analysis, MRLocus provides an estimate of the dispersion of effects around the estimated slope from nearly-LD-independent signal clusters, analogous to PTWAS' use of the $I^2$ statistic for effect size heterogeneity. The combined information from PTWAS and MRLocus regarding both uncertainty in estimation of the gene-to-trait slope, and estimated dispersion or heterogeneity of effects is critical when modeling context-specific (e.g. relevant tissue, cell type, or developmental stage) gene expression as a mediator for downstream traits. Different combinations of eQTL and GWAS SE (primarily influenced by sample size), extent of heterogeneity of effects, and the number of nearly-LD-independent signal clusters within a locus all may give rise to the same gene-to-trait effect and SE, but disentangling these sources of variance is important for experimental planning. For example, consider experimental follow-up for endophenotype downstream traits that could be feasibly measured *in vitro*. An investigator could choose between modulating gene expression directly or modulating the activity of an RE harboring candidate causal SNPs. A nonzero gene-to-trait effect with narrow credible interval estimated by MRLocus (as in Figs 1A and 4B) would suggest that modulating the gene directly or via the activity of an RE should affect the downstream trait, and the predicted effect could be assessed experimentally. However, high dispersion around the gene-to-trait slope (as in Fig 1B) suggests that perturbation of an RE implicated by candidate causal SNPs may induce an effect on the trait that is far from the effect size indicated by the fitted line, such that modulation of the gene directly may prove more successful. MRLocus provides a band around the predominant gene-to-trait slope, such that functional experiments per RE can therefore be prioritized. In all, here we demonstrate the MRLocus method and software utilizing summary statistics from eQTL and GWAS to identify genes mediating traits that can be prioritized for experimental follow-up.

## Supporting information

**S1 Methods. Description of statistical methods used in MRLocus.**
(PDF)

**S1 Text. Supplementary Table and Figures.**
(PDF)

## Acknowledgments

The authors would like to acknowledge the following individuals for helpful comments and suggestions on the work: Karen Mohlke, Laura Raffield, Alaine Broadaway, Robert Gentleman, Steven Munger, Eric Fauman, and Rob Moccia. The authors would like to thank Nicholas Mancuso for assistance in maintaining the 'twas_sim' simulation framework. The authors would like to thank Tobias Strunz and Bernhard Weber for their help in providing summary statistics for the three genes from their liver meta-analysis eQTL study.

The Genotype-Tissue Expression (GTEx) Project was supported by the Common Fund of the Office of the Director of the National Institutes of Health, and by NCI, NHGRI, NHLBI, NIDA, NIMH, and NINDS.

## Author Contributions

**Conceptualization:** Anqi Zhu, Nana Matoba, Joseph G. Ibrahim, Jason L. Stein, Michael I. Love.

**Data curation:** Nana Matoba, Emma P. Wilson, Amanda L. Tapia.

**Formal analysis:** Anqi Zhu, Nana Matoba, Emma P. Wilson, Amanda L. Tapia, Michael I. Love.

**Funding acquisition:** Jason L. Stein, Michael I. Love.

**Methodology:** Anqi Zhu, Nana Matoba, Yun Li, Joseph G. Ibrahim, Jason L. Stein, Michael I. Love.

**Software:** Anqi Zhu, Nana Matoba, Jason L. Stein, Michael I. Love.

**Supervision:** Yun Li, Joseph G. Ibrahim, Jason L. Stein, Michael I. Love.

**Visualization:** Nana Matoba, Emma P. Wilson, Amanda L. Tapia, Jason L. Stein, Michael I. Love.

**Writing – original draft:** Anqi Zhu, Nana Matoba, Jason L. Stein, Michael I. Love.

**Writing – review & editing:** Anqi Zhu, Nana Matoba, Emma P. Wilson, Amanda L. Tapia, Yun Li, Joseph G. Ibrahim, Jason L. Stein, Michael I. Love.

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
