## [Decision Letter · Decision Letter 0]

5 Nov 2020

Dear Dr Love,

Thank you very much for submitting your Research Article entitled 'MRLocus: identifying causal genes mediating a trait through Bayesian estimation of allelic heterogeneity' to PLOS Genetics. Your manuscript was fully evaluated at the editorial level and by independent peer reviewers. The reviewers appreciated the attention to an important problem, but raised some substantial concerns about the current manuscript. Based on the reviews, we will not be able to accept this version of the manuscript, but we would be willing to review again a much-revised version. We cannot, of course, promise publication at that time.

If you decide to revise the manuscript for further consideration at PLOS Genetics, please aim to resubmit within the next 60 days, unless it will take extra time to address the concerns of the reviewers, in which case we would appreciate an expected resubmission date by email to plosgenetics@plos.org.

[LINK]

We are sorry that we cannot be more positive about your manuscript at this stage. Please do not hesitate to contact us if you have any concerns or questions.

Yours sincerely,

Caroline Relton, PhD

Associate Editor

PLOS Genetics

David Balding

Section Editor: Methods

PLOS Genetics

Reviewer's **Comments to the Authors:**

Reviewer #1: Thank you for the opportunity to review this manuscript. The authors have developed a new method to integrate eQTL data with findings from GWAS to help develop insight into the underlying biology responsible for trait-associated genetic variants. This area of research has been extremely busy in the last few years with new methods seemingly appearing every month or so. That said, I believe the MRLocus method devised by these authors is one of the better and innovative approaches in this space and am interested to apply it myself in future work. However, I do have some comments, mainly concerning choices of parameters and requests for further simulations, which I would kindly ask the authors to address before I believe it is suitable for publication.

Comments:

1) The authors state in the introduction that MRLocus estimates gene-to-trait effects at loci containing ‘multiple LD-independent’ eQTL. However, in both their simulations and applied analyses they apply a threshold of r2 < 0.2 (based on the 1KG ref panel). I think this is slightly misleading to the reader, as in the applied analysis of SORT1 it suggests there are 12 ‘independent’ eQTL in liver tissue. However, based on this threshold I think that these 12 variants may well be weakly correlated with each other. In a conventional MR analysis I would be very skeptical of an analysis using this threshold as weakly correlated instruments are likely to result in ‘double counting’ which can bias findings.

I therefore think it is important that the authors run simulations to investigate this, using a more stringent threshold i.e. r2<0.001 to compare how their method compares to other approaches using truly independent instruments. Furthermore, I would like to see how false positive rates are affected when the r2 threshold is varied. My concern is that MRLocus may be prone to false positive findings using an r2 < 0.2, although it will be interesting to see how this compares to other methods in the field. Additionally, the instances of ‘independent’ eQTL should be changes if the authors can demonstrate that their method performs well under the default setting of r2<0.2.

2) Related to my previous point, I think it should be made clear in the applied analysis how many instruments are used at each locus. Although it is available to the reader in the supplementary material, making this clear in the main text means that the reader doesn’t have to seek this information out. The authors highlight SORT1 which had n=12 instruments in their analysis, although for example MRAS only had a single eQTL instrument. Can the authors comment on whether their method provides added value when this is the case (i.e. single instrument MR) over say the eCAVIAR method which their method is built on?

3) I’m also concerned by the P<0.001 threshold applied to identify eQTL as instruments. Can the authors please evaluate how a more stringent threshold influences the false positive rates of their method compared to those it is benchmarked against? In a conventional MR setting an F-stat < 10 is typically used to indicate weak instruments, which will likely be the case when eQTL have P=10-03. This issue is not highlighted in this work as the authors have selected 5 genes which have strong prior knowledge that they are involved in disease risk/trait variation. However, if readers are interested in applying MRLocus to detect novel genes responsible for GWAS signals (which should be its primary purpose I would’ve thought?) then the choice of p-value threshold will likely be important in mitigating false positive rates.

In my opinion applying conventional MR instrument criteria of P<5x10-08 and r2<0.001 is not necessary for cis-QTL analyses, however I think the authors should simulate this to investigate how these parameters influence the performance of their method. Some discussion regarding this would also be welcome for readers interested in applying their approach but perhaps have not studied MR extensively.

4) Although the authors have used the 1KG reference panel is their study for LD calculations and clumping, this dataset consisting of 503 European individuals which based on modern standards is severely outdated. Although I won’t ask the authors to re-run their analyses, I would ask that they mention in their discussion that their method (as with others in the field) would really benefit from larger reference panel for more accurate LD calculations. Even though they have restricted variants to MAF > 0.01, n=503 is going to be unreliable for low frequency SNPs in my experience.

5) The authors do an excellent job of providing a summary of genetic colocalization & TWAS methodologies in their introduction, as well as why introducing MR to address the challenges of deciphering GWAS signals is important. This is a minor suggestion, but I felt that paragraphs 4-6 of the introduction slightly got away from the core message of this paper and may be slightly overwhelming for readers not primarily involved in this topic of research. Perhaps these paragraphs could therefore be streamline, focusing on the methods which are used to benchmark MRLocus, rather than the plethora of other approaches in this very busy area of research.

Reviewer #2: The authors present a new approach to summary statistics-based causal inference tailored to gene expression-to-trait settings, which includes a preprocessing step of colocalization analysis. They have shown across a wide range of simulation settings that their newly proposed method has lower error, less bias and better coverage of the confidence interval than state-of-the-art competing methods. They also apply their method to real data and replicate well-established causal links between certain expression levels and complex traits. While the method seems solid and the simulation results are promising, I suspect that some of the modelling (and simulation) assumptions are unrealistic and may give an unfair advantage to MRclust. The real data application could be more thorough, but it is acceptable for a mainly method paper. Below I list my comments and questions for the authors that may improve the m/s.

Major comments:

It would be clearer if the authors stated the exact mathematical model (in equations) for the simulation process instead of referring to another publication and software (twas_sim). It is not clear for example how much horizontal pleiotropy was added to the signals (i.e. violation of the MR assumption)?

I feel there is a slight confusion between two concepts: expression mediated heritability [A] (which is the fraction of trait heritability mediated by gene expression) vs mediated trait variance explained [B] (as far as I understand from the authors’ definition: the squared standardized (SD/SD) causal effect). For me the relationship between them is A = h2-expression x B, do I understand well? Would be great if the terminology could be clarified in the paper.

While I see the difference between the colocalization step of MRlocus and eCAVIAR, I’m not sure how different results they would give. (Z-scores can be easily transformed back to effect sizes.) Have the authors compared head-to-head their approach to eCAVIAR when applied to the simulated data set to show superior performance?

“For all simulations, if there were no SNPs in the simulated locus with eQTL un-adjusted p-value < 0.001 then a new seed was drawn.” – This leads to Winner’s curse bias, since you only use summary statistics that (by chance) pass the significance threshold, i.e. their effects will be overestimated and hence lead to an underestimation of the causal effect.

Low coverage can be a consequence of two issues: (unaccounted) bias and/or badly calibrated SE. It would be helpful if the authors could explore what is the main reason for low coverage of all methods. To me it stems mostly from bias – can the authors show how methods compare in terms of bias?

Fig 2A shows that MRlocus outperforms the oracle (known causal SNPs) both in terms of RMAE/MAE. The authors argue that it is due to the re-estimation of the effects using colocalization. This however would only work if there is no pleiotropy present at all. We know from real data, that there is almost always pleiotropy present, so simulation setting must be extended to more realistic situations. (E.g. TWMR has an outlier removal step, analogous to MR-PRESSO to protect against such situations) In addition: it seems to me that in the simulations there were no trait-only effects simulated, which is again a very important omission. Gene->trait causal effects do not mean that all trait-associated SNP at the locus must be expression associated, traits can have additional genetic components. Adding this more realistic setting would show the weakness of MRlocus, because it would artificially pull up the effect of those SNPs on expression. Both these situations should be incorporated into the simulations, otherwise the settings are too artificial and unrealistic.

While it is reassuring to see that previous findings are confirmed by MRlocus, but would be nice to see examples of gene-trait associations missed by other methods and only picked up by MRlocus or potentially spurious causal effects (by TWMR or PTWAS) disproved by MRclust.

Since the authors point out the importance of allelic heterogeneity, larger eQTL data sets, such as the eQTL-Gen summary statistics (whole blood, n=32K) would be more ideal to demonstrate the advantages of their method. Have the authors tried MRclust on such data?

I’d move quite some part of the supplementary methods (especially the method description) to the main text Methods – especially given that PLoS Gen has no strict page limit.

Minor comments:

“the original TWMR paper performed conditional analyses” – TWMR performs approximate conditional analysis, which is designed for summary statistics.

“We therefore considered even lower heritability of trait on gene expression of 0.5% and 0.1%” – Please rephrase this sentence, it is currently not understandable, how a trait can have a heritability on expression?

Fig 4B vertical whiskers are missing to indicate SE.

**Have all data underlying the figures and results presented in the manuscript been provided?**

Reviewer #1: Yes

Reviewer #2: Yes

PLOS authors have the option to publish the peer review history of their article (what does this mean?). If published, this will include your full peer review and any attached files.

Reviewer #1: No

Reviewer #2: No

---

## [Decision Letter · Decision Letter 1]

26 Feb 2021

Dear Dr Love,

Thank you for attending to the reviewer's comments so thoroughly. There are a few very minor issues outstanding that have been highlighted by one reviewer (see below). Please address these when preparing your final submission but we are confident that you will address these easily so that we can already decide that your manuscript entitled "MRLocus: identifying causal genes mediating a trait through Bayesian estimation of allelic heterogeneity" has been editorially accepted for publication in PLOS Genetics. Congratulations!

Yours sincerely,

Caroline Relton, PhD

Associate Editor

PLOS Genetics

David Balding

Section Editor: Methods

PLOS Genetics

Comments from the reviewers (if applicable):

Reviewer's Responses to Questions

**Comments to the Authors:**

Reviewer #1: Thank you for addressing my comments and congratulations on an excellent study.

Reviewer #2: The authors have done an excellent job addressing my comments. Only a few minor ones I’d like to flag up:

1. “βjeQTL ~ N(0, (h2g / ncausal )^{-1/2} ).” – I’m not sure why the per SNP variance needs to be raised to the power -1/2?

2. “ygene = Z βeQTL + ε,” Define epsilon in the equations, not only buried in the text. Otherwise the reader doesn’t understand why you introduced s_{err}^2.

3. Please provide the exact equation for y_trait instead (or at least on top) of the ambiguous description in the text.

4. The figures are of very poor resolution, pls change.

**Have all data underlying the figures and results presented in the manuscript been provided?**

Reviewer #1: Yes

Reviewer #2: Yes

PLOS authors have the option to publish the peer review history of their article (what does this mean?). If published, this will include your full peer review and any attached files.

Reviewer #1: **Yes: **Tom Richardson

Reviewer #2: No

**Data Deposition**

http://datadryad.org/submit?journalID=pgenetics&manu=PGENETICS-D-20-01297R1

**Press Queries**

---

## [Editor Report · Acceptance letter]

7 Apr 2021

PGENETICS-D-20-01297R1 

MRLocus: identifying causal genes mediating a trait through Bayesian estimation of allelic heterogeneity 

Dear Dr Love, 

We are pleased to inform you that your manuscript entitled "MRLocus: identifying causal genes mediating a trait through Bayesian estimation of allelic heterogeneity" has been formally accepted for publication in PLOS Genetics! Your manuscript is now with our production department and you will be notified of the publication date in due course.

With kind regards,

Alice Ellingham

PLOS Genetics

On behalf of:
